# Comparative Effects of Flaxseed Sources on the Egg ALA Deposition and Hepatic Gene Expression in Hy-Line Brown Hens

**DOI:** 10.3390/foods9111663

**Published:** 2020-11-14

**Authors:** Muhammad Suhaib Shahid, Tausif Raza, Yuqin Wu, Mazhar Hussain Mangi, Wei Nie, Jianmin Yuan

**Affiliations:** 1State key Laboratory of Animal Nutrition, College of Animal Science and Technology, China Agricultural University, Beijing 100193, China; suhaib_shahid@yahoo.com (M.S.S.); drtraza1@gmail.com (T.R.); yuqin.wu@monash.edu (Y.W.); niewei@cau.edu.cn (W.N.); 2Key Laboratory of Animal Epidemiology and Zoonosis, Ministry of Agriculture, National Animal Transmissible Spongiform Encephalopathy Laboratory, College of Veterinary Medicine, China Agricultural University, Beijing 100193, China; drmazharmangi114@gmail.com

**Keywords:** carbohydrase enzymes, flaxseed oil, extruded flaxseed meal, n-3 eggs, egg quality, β-oxidation genes

## Abstract

Healthy diets are necessary for both humans and animals, including poultry. These diets contain various nutrients for maintenance and production in laying hens. Therefore, research was undertaken to explore the efficiency of various dietary flaxseed sources on the n-3 deposition in the egg yolk and gene expression in laying hens. Five dietary groups were analyzed, i.e., (i) a corn-based diet with no flaxseed (FS) as a negative control (NC), (ii) a wheat-based diet supplemented with 10% whole FS without multi-carbohydrase enzymes (MCE) as a positive control (PC), (iii) ground FS supplemented with MCE (FS), (iv) extruded flaxseed meal was supplemented with MCE (EFM), (v) flaxseed oil supplemented with MCE (FSO). Results indicated that egg weight was highest in the NC, FS, EFM, and FSO groups as compared to PC in the 12th week. Egg mass was higher in enzyme supplemented groups as compared to the PC group, but lower than NC. In the 12th week, the HDEP (hen day egg production) was highest in the FS and EFM groups as compared to FSO, PC, and NC. The FCR (feed conversion ratio) was better in enzyme supplemented groups as compared to the PC group. Enzyme addition enhanced the egg quality as compared to PC in the 12th week. The HDL-C (high-density lipoprotein cholesterol) was increased, while LDL-C (low-density lipoprotein cholesterol), VLDL-C (very-low-density lipoprotein cholesterol), TC (total cholesterol), and TG (total triglycerides) were reduced in the enzyme supplemented groups as compared to PC and NC. The FSO deposit more n-3 PUFA and docosahexaenoic acid (DHA) in the egg yolk as compared to FS and EFM groups. The expression of *ACOX1*, *LCPT1*, *FADS1*, *FADS2*, and *ELOV2* genes were upregulated, while *PPAR-α* was downregulated in the FSO group. The *LPL* mRNA expression was upregulated in the FS, EFM, and FSO groups as compared to the PC and NC groups. It was inferred that FSO with enzymes at 2.5% is cost-effective, improves the hen performances, upregulated the fatty acid metabolism and β-oxidation genes expression, and efficiently deposits optimal n-3 PUFA in the egg as per consumer’s demand.

## 1. Introduction

Healthy diets are necessary for both humans and animals, including poultry. These diets contain various nutrients for maintenance and production in laying hens, yet they need to be carefully formulated in order to reduce harmful substances. The Alpha-Linolenic acid (ALA), also known as omega-3 polyunsaturated fatty acids (n-3 PUFA), has numerous health benefits. Studies on humans and animals demonstrated that n-3 PUFA is conducive to degenerate atherosclerosis and improve brain health [1,2]. Flaxseed (*Linum usitatissimum* L) is broadly used as an n-3 PUFA source to produce n-3 enriched eggs [3]. However, flaxseed contains a large portion of anti-nutritional factors (ANF), such as trypsin inhibitors, myo-inositol phosphate, cadmium, and cyanogenic glycosides [4], which increase the chyme viscosity, thus decreasing its digestibility. Flaxseed supplementation reduced the growth performances in Peking ducks [5], and causes acute colonic mucosal injury, inflammation, and altered cecal microbiota [6]. In laying hens, flaxseed reduces production performances [7] and increase the incidence of liver hemorrhages [8].

Anti-nutritional factors need to be eliminated for a healthy diet by the addition of certain elements. The use of a multi-carbohydrase enzyme can degrade non-starch polysaccharides (NSP) in the flaxseed and enhance n-3 PUFA deposition in eggs [9]. Our previous study showed that MCE supplementation could enhance the n-3 PUFA deposition in eggs [10]. It has been suggested that extruded flaxseed meal (EFM) can minimize the risks associated with anti-nutritional factors of whole flaxseed [11], and help to produce eggs with ≥ 300 mg per egg n-3 PUFA [12]. In addition to EFM, flaxseed oil (FSO) is another alternative source of n-3 PUFA for egg enrichment [13]. Although, some reports comparing FSO with milled flaxseed argued that fatty acid deposition from flaxseed oil was two times greater as compared to milled flaxseed when fed at the same dietary inclusions [3]. Notably, the addition of exogenous enzymes in milled flaxseed-based diets was not evaluated in their experiment.

Laying hens can efficiently absorb n-3 PUFA from the diet and transfer and deposit it into the yolk [14]. It is well known that several enzymes are involved in lipogenesis [15], and dietary fatty acid can influence the expression of a gene coding for these enzymes [16], such as lipoprotein lipase (LPL) [17]. Moreover, the conversion of ALA to long-chain n-3-PUFA involves a series of elongation and desaturation steps, catalyzed by enzymes, such as elongase of very-long-chain fatty acids (ELOV2 and ELOV5) and fatty acid desaturase 1 and 2 (FADS1 and FADS2) [18]. The Acyl-coenzyme A oxidase 1 (ACOX1), and liver carnitine-palmitoyl-transferase 1 (L-CPT1) are involved in the oxidation of fatty acids [19].

In the current study, grounded flaxseed or extruded flaxseed meal with multi-carbohydrase enzyme supplementation and purified extracted flaxseed oil were fed to hens. Performance, egg quality, fatty acids deposition in egg, serum lipid metabolites, and hepatic genes expression were measured to explore which form of ALA source is better for the fatty acid transfer from diet to the yolk. This research can evaluate important aspects of feed formulation and expulsion of anti-nutritional factors from the poultry diet.

## 2. Material and Methods

### 2.1. Bird Management

This study was approved by the Animal welfare and Ethical Committee for Laboratory Animals, China Agricultural University, College of Animal Science and Technology (No. AW04110202-2-2), and all experiments were performed according to the guidelines for the care and use of laboratory animals. Five dietary groups (NC, PC, FS, EFM, and FSO) were allocated with a total of 450 hens (38 weeks old) of Hy-line brown breed. Each group was replicated six times, and each replicate had 15 hens, with separate compartments for three hens in each cage. However, five conjoint cages were regarded as the same replicate as fed with the same diet. Birds were housed in an automatically controlled house. Light, feed, and water were given according to the laying hen’s standard requirements during the 12 weeks experimental period. The hens did not receive any vaccines or drugs during the entire experimental period.

### 2.2. Diet Preparation

Flaxseed was imported from Canada, EFM, and flaxseed oil was bought from Zhangjiakou (Hebei, China) local oil processing plant. The corn-based diet was used in negative control with no flaxseed (NC). The wheat-based diet was supplemented with 10% whole flaxseed without enzyme (PC). In the third group, ground FS was supplemented with multi-carbohydrase enzyme (FS). It contains xylanase 35,000, neutral protease 10,000, β-mannanase 1500, β-glucanase 2000, cellulose 500, amylase 100, and pectinase 10,000 (U g^−1^). The EFM group used an extruded flaxseed meal supplemented with MCE. The FSO treatment used flaxseed oil with MCE. The diets were formulated to be iso-caloric and iso-nitrogenous and to meet or exceed requirements for laying hens [20]. All the diets were fed to the hens for a period of 12 weeks as shown in Table 1.

### 2.3. Flock Performance

Eggs were collected and weighed daily to measure egg mass and egg production. Hen body weight was measured at the beginning and then after every two weeks interval. Feed consumed every two weeks was also recorded to calculate the body weight gain and other performances. Hen day egg production was recorded on daily basis, and calculated every two weeks. FCR was calculated based on egg mass.

### 2.4. Egg Quality and Fatty Acid Profile

At the start of the 4th week, three eggs from each replicate were collected to measure the egg quality. This was repeated every two weeks till the 12th week. The albumen height and Haugh unit were analyzed by using Digital EGG TESTER (DET-6000). From the end of 2nd week, the eggs were collected for FA analysis. The egg yolks were separated, rolled on filter paper, pooled, freeze-dried at −80 °C, and stored at −80 °C for FA analysis. In short fatty acid methyl esters (FAMEs) of egg yolks were prepared using direct FAME synthesis [21]. The FAMEs were subjected to gas chromatography-mass spectrometry (SCION-456, China) for fatty acid separation. The column measurements were (30 m × 0.32 mm × 0.25 m). The carrier gas used was helium at a flow rate of 1.5 cm 3/min. The column, detector, and injector temperatures were set at 195, 250, and 225 °C, respectively. FAs were identified comparing their retention times with authentic standards.

### 2.5. Serum Biochemical Profile

Blood was taken from the wing vein of one bird per replicate at the end of the 4th, 6th, 8th, 10th, and 12th week. Blood was centrifuged at 4000 G for 15 min. The serum was separated and stored at −80 °C for further analysis. The serum indices, namely TC, TG, VLDL, LDL-C, HDL-C were measured using commercially available kits (Nanjing Jiancheng Bioengineering Institute, China) following the manufacturer’s instructions. Chicken lipoprotein lipase (LPL) was measured by ELISA Kit (DRE-C9561b, Shanghai, China).

### 2.6. RNA Extraction and Reverse Transcription

At 12 weeks, one bird from every replicate was slaughtered by cutting the jugular vein. The liver was removed and stored at −80 °C for the analysis of gene expression. The gene (Table 2) mRNA expression was measured as described by Shin et al. [22]. Total RNA of liver samples was extracted using Trizol reagent (Invitrogen Biotechnology Inc., Carlsbad, CA, USA) according to the manufacturer’s protocol. The primers were designed through NCBI Primer Blast. Results of relative mRNA expression genes were calculated using the 2^−ΔΔCt^ method [23].

### 2.7. Statistical Analysis

Data were analyzed using a completely randomized design by SPSS 20.0 [24]. Means were compared with one-way ANOVA with a significance level of *p* < 0.05. The least significant difference test (LSD) was performed using Tukey’s b-test. Two-way ANOVA was performed to show whether interaction exists between groups and weeks of the experiment. The Tukey’s b-test was used to compare the means. The principal component analysis (PCA) test was used to highlight the differences among the dietary groups.

## 3. Results

### 3.1. Performance

The sources of flaxseed affected the performance of laying hens (Table 3). Compared to the control (NC), supplemental 10% flaxseed (PC) did not affect egg weight in the first six weeks but significantly decreased in the 8th week. In the same group, there was a decrease in egg mass, hen day egg production, and feed conversion ratio of hens from 10 weeks. On the contrary, supplemental 10% flaxseed (PC) improved feed intake in the 10th week (*p* < 0.05) as compared to other supplemental groups.

Compared to the 10% flaxseed (PC), supplemental 10% flaxseed with enzymes (FS) significantly improved egg weight, egg mass, hen day egg production, feed intake, and feed conversion ratio in the 10th week. The egg weight was similar in the FS and NC groups, however egg mass in the FS group was lower than the NC group.

As compared to the 10% flaxseed with enzymes (FS), the egg weight was similar in EFM, FSO, and NC groups. The egg mass was significantly similar in FS, EFM, and FSO, however, it was lower than the NC group. The HDEP during the 10th and 12th weeks was significantly higher in the FS and EFM followed by FSO and NC groups.

Compared to 10% flaxseed (PC), the FS, EFM, FSO, and NC significantly improved FCR in the 10th week. At 12 weeks, better FCR was reported in the NC group, followed by FS, EFM, and FSO groups while the highest FCR was reported in the PC group.

There was an interaction for egg weight between diet and weeks, where the FSO group during the 10th week had the highest egg weight. For egg mass, there was also an interaction between groups and weeks, where the NC group from week 4 to 12 had the highest egg mass. There was no interaction for HDEP. However, for groups and weeks, there was a significant effect on egg production. The HDEP was highest for NC, FS, EFM, and FSO groups from 4 to 12 weeks. There was no interaction for BWG. However, for groups and weeks, there were significant interactions for BWG where the PC group at the 8th and 10th weeks had the highest BWG. There was no interaction between groups and weeks for FI, however, for groups and weeks, there were significant effects, where the PC group at 10th and 12th weeks had the highest FI. There was an interaction for FCR where PC had the highest FCR. The PCA analysis showed that NC and PC showed more variation as compared to FS, EFM, and FSO groups for egg weight, egg mass, HDEP, BWG, FI, and FCR, as shown in Appendix A.

### 3.2. Egg Quality

The n-3 sources of flaxseed affected egg quality (Table 4). Compared to the control (NC), supplemental 10% flaxseed (PC) tended to decrease albumen height in the 8th week and significantly decreased albumen height of eggs in the 12th week, and significantly decreased Haugh unit of the egg white from the 8th to 12th week (*p* < 0.05).

Compared to the 10% flaxseed (PC), supplemental 10% flaxseed with enzymes (FS) improved albumen height and Haugh unit from 8 to 12 weeks.

Compared to 10% flaxseed with enzymes (FS), supplemental EFM decreased the albumin height in the 8th and 10th weeks, but it was significantly similar to FS at 12 weeks. Haugh unit was significantly similar in 8th, 10th, and 12th weeks in FS and EFM groups.

Compared to 10% flaxseed with enzymes (FS), supplemental EFM and FSO decreased albumin height at 8th and 10th weeks, but it was significantly similar to FS at 6th and 12th weeks. The Haugh unit was significantly similar in FS, EFM, FSO, and NC in the 8th and 10th weeks. However, in the 12th week, the Haugh unit was lower in PC as compared to FS, EFM, and FSO groups.

A significant interaction between dietary groups and weeks was reported for albumen height where, FS, EFM, and FSO groups reported the highest values of albumen height. There was no interaction between diets and weeks for the Haugh unit. However, diet and weeks had a significant effect on the Haugh unit where it was highest in NC, FS, EMF, and FSO groups during the 6th and 8th weeks. The PCA analysis for egg quality traits showed more variation in the NC and PC groups as compared to the other treatment groups (Appendix A). 

### 3.3. Serum Lipid Metabolites

The dietary treatments affected the serum lipid profile (Table 5). In the 4th, 6th, 8th, 10th, and 12th weeks, TC was highest in the NC, which was followed by PC, while the lowest was recorded in the FS, EFM, and FSO groups.

In the 4th, 6th, 10th, and 12th weeks, TG was highest in NC, which was followed by PC while the lowest was recorded in the FS, EFM, and FSO. In the 8th week, TG was highest in NC and PC while lowest in FS, EFM, and FSO.

In the 4th and 8th weeks, LDL-C was reported highest in the NC group, while the lowest was reported in the PC, FS, EFM, and FSO groups. In the 6th, 10th, and 12th weeks, the highest LDL-C was recorded in NC which was followed by PC, while the lowest was reported in FS, EFM, and FSO.

In the 4th, 6th, 8th, and 10th weeks, HDL-C was lowest in the NC, while highest in the PC, FS, EFM, and FSO groups, respectively. In the 12th week, HDL-C was lowest in the NC, followed by PC, EFM, and FSO groups, while the highest was reported in the FS group.

In the 4th, 6th, 8th, 10th, and 12th weeks, VLDL-C was highest in the NC group, followed by PC, FS, and EFM groups, while the lowest value was noted in the FSO group, respectively.

The serum LPL was highest in the PC, FS, EFM, and FSO groups, while lowest in the NC group in the 4th, 6th, 8th, 10th, and 12th weeks.

There was no interaction for TC, TG, LDL-C, HDL-C, VLDL-C, and LPL. The groups and weeks alone showed a significant difference. The groups showed a significant difference for TC and TG where the lowest TC was reported in the FS, EFM, and FSO groups. The weeks showed a significant difference among the weeks where from 2 to 6 weeks had the lowest TC and from 2 to 4 weeks TG was lowest. Among the groups, FS, EFM, and FSO had the lowest LDL-C, whereas FS had the highest HDL-C among the groups. The lowest VLDL-C was significantly highest for the FSO group. For LPL the highest values among the groups were PC, FS, EFM, and FSO groups. The PCA test demonstrated that NC had the highest variation scale as compared to other groups (Appendix A).

### 3.4. Fatty Acid Profile of Egg (mg/g)

The sources of flaxseed affected the fatty acid profile of egg yolk (Table 6). In the 2nd week, compared to the control (NC), supplemental 10% flaxseed significantly improved DHA and total n-3 PUFA contents of the egg, and decreased total n-6 and n-6/n-3 (*p* < 0.05). Compared to PC, supplemental 10% flaxseed with enzymes (FS), significantly improved the DHA and total n-3 PUFA content of the egg, and decreased n-6/n-3 from 2nd to the end of the trial. From the 2nd to 12th week, EFM had significantly similar DHA to the FS group. In the 2nd week, DHA was similar in the FS, EFM, and FSO groups. Compared to the FS and EFM groups, the DHA was improved from the 4th week till the end of the trial in the FSO group. From 2nd to 8th weeks, the total n-3 PUFA was significantly similar in the FS, EFM, and FSO groups. In the 10th and 12th weeks, the FSO deposited the highest total n-3 PUFA as compared to FS and EFM. From 2nd to 12th weeks, the ratio of n-6:n-3 was improved in the FS, EFM, and FSO groups as compared to the PC and NC groups.

There was an interaction between groups and weeks for DHA, where FSO at 8th, 10th, and 12th weeks had the highest DHA values. For total n-6 PUFA, there was an interaction between groups and weeks, where EFM and FSO groups at 8th,10th, and 12th weeks had the lowest total n-6. There was also an interaction between groups and weeks where FSO at 10th and 12th weeks had the highest total n-3. An interaction was found between groups and weeks where the ratio of n-6 to n-3 PUFA was lowest in EFM and FSO groups at 10th and 12th weeks. The PCA test showed that most variation occurred in the NC group (Appendix A).

### 3.5. Hepatic Gene Expression

The n-3 sources of flaxseed affected the hepatic gene expression (Table 7). The mRNA expression of *ACOX1, LCPT1, PPAR-α, FADS1, FADS2, ELOV2,* and *LPL* was significantly (*p* < 0.05) affected by the treatments. The expression of *ACOX1, LCPT1, FADS1, FADS2,* and *ELOV2* genes was upregulated in the FSO group, which was followed by EFM and FS groups while it was lowest in the NC and PC groups. The mRNA expression of *PPAR-α* was upregulated in the NC, which was followed by PC, FS, EFM groups while downregulated in the FSO group. The *LPL* mRNA expression was upregulated in the FS, EFM, and FSO groups while it was downregulated in the NC and PC groups.

## 4. Discussion

Flaxseed is a well-known source of n-3 PUFA, however, the presence of water-soluble NSP as the major ANF in the cell wall limits its use in poultry feeding [25]. In our study, the flaxseed fed group (PC) reduced the egg weight and egg mass, whereas increased the feed intake and FCR as compared to other groups. The poor performance of hens fed FS alone might be due to the ANF like NSP and cyanogenic glycosides [26]. These compounds increase the viscosity and decrease the nutrients digestibility and ultimately limit the nutrient’s availability for hen’s better performance [25,27]. Poor nutrient utilization can also lead to poor tissue n-3 FA deposition [8]. Moreover, flaxseed increased the cecal microbiota causing inflammation in ducks [6]. Our results agreed with Mridula et al. [28], who also observed poor FCR in broilers fed with flaxseed. Contrary to our findings, Ehr et al. [3] reported decreased performance of broilers fed milled flaxseed and FSO without a carbohydrase enzyme.

In this study, the supplementation of a multi-carbohydrase enzyme with FS, EFM, and FSO alleviates the ANF of flaxseed leading to better performance. Similarly, Moghadam et al. [29] reported better performance of hens when flaxseed was supplemented with a blend of carbohydrase enzymes. Supplementing flaxseed for a longer duration decreased the performance of hens at the 10th and 12th weeks of the experiment in the PC group, while the NC, FS, EFM, and FSO enhanced the performance of hens. The enhanced performance of hens in the FS, EFM, and FSO might be due to the lower NSP in these respective groups. The multi-carbohydrase enzymes contain a variety of activities and their effect in degrading cell walls and releasing nutrients for utilization may be synergistic [30]. The use of multi-carbohydrase enzymes with flaxseed-based diets can eliminate the anti-nutritive effects of NSP and enhance the n-3 PUFA deposition in the tissue [9].

Extrusion of flaxseed is another method used to eliminate the ANFs. The use of extruded flaxseed meal in poultry stabilizes the viscosity and improves the digestive tract morphology, thus increasing absorptive and digestive activities for better bird performance [31].

The blend of carbohydrase enzymes enhanced the albumen height and Haugh unit in the FS, EFM, and FSO groups as compared to the PC group in which FS was supplemented alone. This might be due to the fact that flaxseed has a good amount of phytoestrogens like lignans and iso-flavones which play an important role in regulating reproductive performance and egg quality [32]. The addition of enzymes to the diet may have enhanced the availability of phytoestrogen leading to an increased egg quality in FS, EFM, and FSO groups. Better egg quality was also reported after supplementing enzyme with FS [9].

The LPL is a lipogenic enzyme that regulates the body’s lipid uptake by hydrolyzing triacylglycerol (TAG) [33]. Unlike mammals, the lipids deposited in the fat tissues of birds are mainly synthesized in the liver and absorbed by the LPL [34]. A direct relationship between LPL activity and TG breakdown existed under different conditions [35]. The increased LPL activity in the serum and higher *LPL* mRNA expression in the liver reduces the TG in the serum of broiler chickens [36]. In our study, higher LPL enzyme activity in the serum and higher hepatic mRNA expression could reduce the TG in the serum of hens. In agreement with this study, flaxseed oil increased the serum HDL-C, while decreased the LDL-C, TGs, and VLDL-C [37].

The biosynthesis of long-chain n-3 PUFA from C18:3 includes a series of desaturation, elongation, and, ultimately β-oxidation, reactions [38]. The *FADS2* or delta-6 desaturase enzyme is vital for the conversion of linoleic acid (C18:2) to arachidonic acid (C20:4), as well as, linolenic acid (C18:3) to DHA (C20:5) and EPA (C22:6). The competition between C18:2 and C18:3 FA for the delta-6-desaturase enzyme is mandatory for elongation of n-3 PUFA and n-6 PUFA to their long-chain metabolites [39]. In our study, overexpression of *FADS1*, *FADS2,* and *ELOV2* genes was observed in the FSO group as compared to FS and EFM. The higher expression of desaturation and elongation genes demonstrate better conversion of C18:3 to DHA and EPA, which was clearly observed in FSO fed hens eggs. The reduction of total n-6 PUFA and enrichment of total n-3 PUFA in the eggs fed FSO also represent elongation and desaturation of n-3 fatty acid. A previous study reported that the n-3 enrichment efficiency of FSO was two times greater than the grounded FS at the same inclusion levels [3]. Unlike our results, the supplementation of flaxseed and chia seed resulted in decreased *FADS1* and *FADS2*, gene expression [40]. Our previous study increased the expression of *FADS2* gene expression when flaxseed was given to ducks [5]. Similarly, Mirshekar et al. [16] reported that flaxseed oil increased the expression of the *FADS2* gene in broilers. The substrate competition for existing enzymes could contribute to the dietary regulation of long-chain n-3 PUFA synthesis [41].

The Carnitine palmitoyl-transferase (*CPT1*) is regarded as the main regulatory enzyme in fatty acid β-oxidation catalyzing the conversion of cytosolic fatty acyl-CoA to fatty acyl-carnitine for entry into mitochondria [42]. The peroxisome proliferator-activated receptor alpha (*PPARα*) can regulate and modulate the expression of genes encoding several mitochondrial and peroxisomal fatty acid β-oxidation enzymes [43,44]. In our study, the highest LCPT1 was reported in the FSO group, indicating higher fatty acid β-oxidation, however, its transcriptional regulator *PPAR-α* was reduced. The findings of this study were supported by Jin et al. [45] who reported higher expression of *LCPT1* and reduced *PPAR-α* expression when different DHA/EPA ratios were given to the black seabream fish. Similarly, Sanz et al. [46] found that the inclusion of sunflower oil in the diet of broiler chickens enhanced the activities of LCPT1. The acyl-coenzyme-A oxidase (*ACOX1*) is the rate-limiting enzyme for this peroxisomal fatty acid β-oxidation [47]. The last phase in the DHA pathway synthesis occurs in peroxisomes through β-oxidation [38]. The readily available ALA in the FSO incorporates more ALA in the liver of hens which ultimately upregulate the *ACOX1* gene expression, leading to the increased formation of DHA in the eggs fed the FSO, compared with other n-3 treatments fed to hens i.e., FS and EFM. Likewise, Li et al. [48] reported the upregulation of *ACOX1* gene expression. The CPT1 is often co-regulated with *ACOX1* [19], which is also demonstrated in this study.

## 5. Conclusions

The addition of enzymes to the FS, EFM, and FSO enhanced the laying performances, egg quality, and serum lipid profiles of hens when compared with the NC group. Among the three n-3 treatments, flaxseed oil with multi-carbohydrase enzymes efficiently deposits more n-3 PUFA and DHA in the egg yolk and upregulated the expression of fatty acid metabolism (*FADS1, FADS2, ELOV2*) genes. Moreover, we found a mechanism that FSO upregulates the β-oxidation genes expression *LCPT1* and *ACOX1* by downregulating their transcriptional regulator *PPAR-α*, as a result, more DHA is deposited in the FSO group. Hence, FSO is cost-effective at 2.5% as it deposited more n-3 PUFA and DHA as compared to FS and EFM at 10%. It is recommended that multi-carbohydrase enzymes should be supplemented to the FSO diet to alleviate the ANF for better performances of hens and optimal n-3 PUFA deposition in the egg for health-conscious consumers. 

## Figures and Tables

**Table 1 foods-09-01663-t001:** Ingredient composition of diets.

Ingredients	NC	PC	FS	EFM	FSO
Corn	62.3				
Wheat		70.7	70.7	69.59	73.6
Soybean meal	25	7.9	7.9	9	11.8
Flaxseed		10	10		
Extruded flaxseed meal				10	
Soybean oil	1.4				
Flaxseed oil					2.5
Limestone	9.5	9.4	9.4	9.4	9.4
Calcium hydro-phosphate	1	0.81	0.8	0.8	0.82
Salt	0.35	0.35	0.35	0.35	0.35
Choline chloride	0.1	0.1	0.1	0.1	0.1
Minerals	0.2	0.2	0.2	0.2	0.2
DL-Methionine	0.11	0.13	0.12	0.12	0.12
L-Lysine		0.36	0.36	0.37	0.35
Vitamin	0.02	0.02	0.02	0.02	0.02
Phytase	0.02	0.02	0.02	0.02	0.02
Compound enzyme			0.02	0.02	0.02
Zeolite powder					0.69
Pigment		0.01	0.01	0.01	0.01
Total	100	100	100	100	100
Nutrient analysis					
AME ((kcal kg^−1^)	2740	2740	2740	2742	2740
Protein %	15.99	15.99	15.99	16.0	15.94
Ca%	3.91	3.91	3.91	3.91	3.91
Available phosphorus%	0.26	0.26	0.26	0.26	0.26
Lysine %	0.79	0.79	0.80	0.81	0.80
Methionine %	0.37	0.38	0.38	0.38	0.37
M+C	0.63	0.63	0.63	0.63	0.62
Fatty acids %					
Palmitic acid C16:0	13.5	9.6	9.4	9.3	9.2
Stearic acid C18:0	17.60	13.93	14.03	13.24	13.32
Oleic acid C18:1	29.6	24.08	24.11	23.27	24.52
Linoleic acid C18:2n6	36.34	43.21	42.63	42.21	43.43
Alpha Linolenic acid n3	8.32	22.56	23.22	23.46	23.75

NC = negative control, PC = positive control, FS = flaxseed, EFM = extruded flaxseed meal, FSO = flaxseed oil.

**Table 2 foods-09-01663-t002:** Sequences of primer pairs of mRNA.

Primers	Primer Sequence 5′–3′	Accession Number
*FADS1*	F: GAGCCATCGGTGAGGGTTTC	XM 420557
	R: CTCCAGTCCTTTCCTTGCGT	
*FADS2*	F: ACTGGTGGAACCATCGTCAC	NM_001160428.2
	R: GCAGAGGTGGGAAGATGAGG	
*ELOV2*	F: CACCGTCGCATACCTGCTCTG	NM_001197308.1
	R: AGGTTCTGGCACTGCAAGTTGTAG	
*ELOV5*	F: GCGATGCGTCCTTATCTGTGGTG	NM_001199197.1
	R: GCTGGTCTGGAAGATTGTCAGGAC	
*LPL*	F: GAC AGC TTG GCA CAG TGC AA	NM_205282.1
	R: CAC CCA TGG ATC ACC ACA AA	
*PPAR-α*	F: AGGCCAAGTTGAAAGCAGA	NM_001001464.1
	R: GTCTTCTCTGCCATGCACAA	
*ACOX1*	F: CCAGTCAGCTTGGTAGAGGC	NM_001006205.1
	R: AGTGACAGTGTGCCTCAGATG	
*LCPT1*	F: GCCCTGATGCCTTCATTCAA	AY675193
	R: ATTTTCCCATGTCTCGGTGA	
*Beta-actin*	R: AAATCAAGATCATTGCCCCACCT	NM_205518.1
	F: AGGGGTGTGGGTGTTGGTAA	

FADS1: fatty acid desaturase 1 encoding ∆-5 desaturase; FADS2: fatty acid desaturase 2 encoding ∆-6 desaturase; ELOV2: elongase 2; ELOV5: elongase 5; PPAR-α: peroxisome proliferator activated receptors alpha; LPL; lipoprotein lipase; ACOX1: acyl-coenzyme A oxidase 1; LCPT1: liver carnitine palmitoyl-transferase 1.

**Table 3 foods-09-01663-t003:** Effect of different forms of flaxseed on the performance of Hy-line brown.

Groups								*p*-Values		
	Weeks	NC	PC	FS	EFM	FSO	Mean	Groups	Weeks	Groups × Weeks
**Egg** **weight (g)**	**2**	65.54 ± 0.48	64.64 ± 0.59	64.64 ± 0.29	65.43 ± 0.16	65.99 ± 0.31	65.24 ^AB^	<0.001	<0.001	0.026
	**4**	64.21 ± 0.48	63.59 ± 0.71	63.76 ± 0.29	64.79 ± 0.38	65.01 ± 0.32	64.27 ^C^			
	**6**	64.66 ± 0.34	64.64 ± 0.66	64.42 ± 0.16	65.02 ± 0.40	64.83 ± 0.36	64.72 ^BC^			
	**8**	65.11 ± 0.34 ^a^	63.95 ± 0.38 ^b^	64.06 ± 0.20 ^b^	64.86 ± 0.46 ^ab^	65.68 ± 0.19 ^a^	64.73 ^BC^			
	**10**	65.86 ± 0.28 ^a^	64.15 ± 0.18 ^b^	66.06 ± 0.37 ^a^	66.18 ± 0.30 ^a^	65.83 ± 0.29 ^a^	65.62 ^A^			
	**12**	65.87 ± 0.39 ^a^	63.22 ± 0.44 ^b^	65.88 ± 0.21 ^a^	65.76 ± 0.31 ^a^	65.93 ± 0.23 ^a^	65.33 ^AB^			
**Mean**		65.21 ^AB^	64.03 ^C^	64.80 ^B^	65.34 ^AB^	65.54 ^A^				
**Egg** **mass (g)**	**2**	59.41 ± 0.70	59.06 ± 0.87	57.76 ± 0.92	58.12 ± 0.99	59.2 ± 0.83	58.71 ^B^	<0.001	<0.001	<0.001
	**4**	60.20 ± 0.76	60.37 ± 1.17	59.42 ± 0.67	60.05 ± 0.40	60.16 ± 0.39	60.04 ^A^			
	**6**	60.80 ± 0.86	59.86 ± 1.07	59.74 ± 0.66	59.38 ± 1.33	60.25 ± 1.03	60.00 ^A^			
	**8**	60.94 ± 0.87	59.87 ± 1.01	60.83 ± 0.91	60.36 ± 0.48	60.57 ± 0.64	60.51 ^A^			
	**10**	63.01 ± 0.26 ^a^	56.26 ± 0.25 ^c^	61.21 ± 0.24 ^b^	61.59 ± 0.63 ^b^	61.93 ± 0.69 ^ab^	60.80 ^A^			
	**12**	63.64 ± 0.47 ^a^	54.11 ± 0.22 ^c^	60.94 ± 0.30 ^b^	61.53 ± 0.38 ^b^	60.99 ± 0.31 ^b^	60.24 ^A^			
**Mean**		61.33 ^A^	58.25 ^C^	59.98 ^B^	60.17 ^B^	60.51 ^AB^				
**Hen** **day** **egg** **production (%)**	**2**	91.19 ± 2.00	88.33 ± 1.63	92.22 ± 1.10	89.76 ± 2.07	91.67 ± 1.78	90.63 ^B^	<0.001	<0.001	0.119
	**4**	94.05 ± 1.48	93.22 ± 0.75	94.28 ± 1.03	93.81 ± 0.77	94.29 ± 1.06	93.93 ^A^			
	**6**	94.29 ± 1.57	93.33 ± 1.68	94.52 ± 1.45	93.69 ± 2.36	94.28 ± 1.61	94.02 ^A^			
	**8**	94.52 ± 1.65	95.12 ± 1.16	95.12 ± 1.26	93.69 ± 2.03	94.52 ± 1.65	94.59 ^A^			
	**10**	95.08 ± 0.92 ^b^	92.02 ± 0.38 ^c^	97.47 ± 0.45 ^a^	97.65 ± 0.38 ^a^	95.66 ± 0.38 ^b^	95.58 ^A^			
	**12**	96.08 ± 0.41 ^b^	88.95 ± 0.74 ^c^	98.04 ± 0.20 ^a^	97.79 ± 0.34 ^a^	95.59 ± 0.24 ^b^	95.29 ^A^			
**Mean**		94.20 ^A^	91.83 ^B^	95.28 ^A^	94.40 ^A^	94.34 ^A^				
**Body** **weight** **gain (g)**	**2**	0.03 ± 0.01	0.02 ± 0.01	0.02 ± 0.01	0.03 ± 0.01	0.03 ± 0.01	0.025 ^B^	<0.045	<0.001	0.063
	**4**	0.02 ± 0.01	0.1 ± 0.07	0.02 ± 0.01	0.04 ± 0.01	0.02 ± 0.01	0.039 ^B^			
	**6**	0.01 ± 0.01	0.03 ± 0.01	0.02 ± 0.01	0.04 ± 0.01	0.03 ± 0.01	0.026 ^B^			
	**8**	0.13 ± 0.03	0.08 ± 0.03	0.15 ± 0.06	0.05±0.02	0.06 ± 0.01	0.095 ^A^			
	**10**	0.1 ± 0.01	0.17 ± 0.06	0.12 ± 0.03	0.09 ± 0.02	0.05 ± 0.01	0.106 ^A^			
	**12**	0.03 ± 0.01	0.03 ± 0.01	0.05 ± 0.02	0.02 ± 0.01	0.02 ± 0.01	0.031 ^B^			
**Mean**		0.53 ^AB^	0.73 ^A^	0.64 ^AB^	0.46 ^AB^	0.34 ^B^				
**Feed intake (g)**	**2**	122.19 ± 2.01	122.36 ± 0.84	120.50 ± 0.54	120.17 ± 1.28	121.09 ± 0.77	120.86 ^C^	<0.000	<0.000	0.854
	**4**	123.17 ± 0.73 ^a^	123.78 ± 0.41 ^a^	120.84 ± 0.23 ^b^	121.20 ± 0.45 ^b^	121.46 ± 0.64 ^b^	122.09 ^B^			
	**6**	122.66 ± 0.45	123.81 ± 0.49	121.11 ± 0.81	121.95 ± 0.63	122.66 ± 0.83	122.44 ^AB^			
	**8**	122.37 ± 0.54	123.39 ± 0.65	121.25 ± 0.68	121.29 ± 0.42	122.33 ± 0.84	122.12 ^B^			
	**1**	123.94 ± 0.53 ^b^	125.36 ± 0.36 ^a^	122.36 ± 0.56 ^c^	122.30 ± 0.32 ^c^	122.78 ± 0.53 ^bc^	123.35 ^A^			
	**12**	123.44 ± 0.85 ^b^	126.57 ± 0.35 ^a^	122.05 ± 0.48 ^b^	122.89 ± 0.50 ^b^	122.22 ± 0.55 ^b^	123.43 ^A^			
**Mean**		122.63 ^B^	124.21 ^A^	121.35 ^C^	121.63 ^BC^	122.09 ^BC^				
**feed** **conversion** **ratio (%)**	**2**	2.02 ± 0.04	2.07 ± 0.02	2.09 ± 0.03	2.07 ± 0.02	2.05 ± 0.03	2.06	<0.001	0.078	<0.001
	**4**	2.05 ± 0.03	2.05 ± 0.05	2.03 ± 0.02	2.02 ± 0.01	2.02 ± 0.01	2.04			
	**6**	2.02 ± 0.03	2.07 ± 0.04	2.03 ± 0.02	2.06 ± 0.05	2.04 ± 0.03	2.04			
	**8**	2.01 ± 0.03	2.06 ± 0.03	2.00 ± 0.02	2.01 ± 0.01	2.02 ± 0.02	2.02			
	**10**	1.97 ± 0.01 ^b^	2.23 ± 0.01 ^a^	2.00 ± 0.01 ^b^	1.99 ± 0.02 ^b^	1.98 ± 0.02 ^b^	2.03			
	**12**	1.94 ± 0.02 ^c^	2.34 ± 0.01 ^a^	2.00 ± 0.01 ^b^	2.00 ± 0.01 ^b^	2.01 ± 0.01 ^b^	2.06			
**Mean**		2.00 ^B^	2.14 ^A^	2.03 ^B^	2.04 ^B^	2.02 ^B^				

NC = negative control, PC = positive control, FS = flaxseed, EFM = extruded flaxseed meal, FSO = flaxseed oil. *n* = 6. Means having similar letters in a row for individual parameters do not differ significantly from one another (*p* > 0.05). The lowercase letters are for one-way Anova represents weekly analysis. The uppercase letters are for two-way Anova represent the grand mean of groups and weeks.

**Table 4 foods-09-01663-t004:** Effect of different forms of flaxseed on the egg quality of Hy-line brown hens.

		Groups						*p*-Value		
Weeks		NC	PC	FS	EFM	FSO	Mean	Groups	Weeks	Groups × Weeks
**Albumen** **height (mm)**	**4**	7.92 ± 0.20	8.1 ± 0.15	7.89 ± 0.25	7.98 ± 0.15	7.95 ± 0.28	7.97	<0.001	0.204	0.002
	**6**	6.82 ± 0.57 ^b^	6.30 ± 0.31 ^b^	8.09 ± 0.41 ^a^	8.40 ± 0.37 ^a^	8.31 ± 0.42 ^a^	7.58			
	**8**	7.00 ± 0.34 ^bc^	6.56 ± 0.38 ^c^	9.06 ± 0.36 ^a^	8.07 ± 0.39 ^ab^	7.99 ± 0.27 ^ab^	7.74			
	**10**	7.62 ± 0.41 ^bc^	6.83 ± 0.37 ^c^	8.74 ± 0.25 ^a^	8.37 ± 0.30 ^ab^	8.70 ± 0.33 ^a^	8.05			
	**12**	7.99 ± 0.23 ^a^	6.00 ± 0.42 ^b^	8.3 ± 0.36 ^a^	8.55 ± 0.26 ^a^	8.54 ± 0.42 ^a^	7.88			
	**Mean**	7.47 ^B^	6.76 ^C^	8.42 ^A^	8.27 ^A^	8.30 ^A^				
**Haugh** **unit (AA)**	**4**	88.05 ± 1.12	87.63 ± 1.25	91.41 ± 0.54	87.73 ± 1.16	87.25 ± 1.48	88.41 ^AB^	<0.001	0.002	0.240
	**6**	93.18 ± 1.86 ^a^	87.13 ± 2.43 ^ab^	92.88 ± 2.44 ^a^	91.97 ± 1.86 ^ab^	91.74 ± 2.26 ^b^	91.38 ^A^			
	**8**	92.58 ± 1.50 ^a^	82.57 ± 1.84 ^b^	92.41 ± 2.07 ^a^	92.31 ± 2.77 ^a^	95.03 ± 1.80 ^a^	90.98 ^A^			
	**10**	91.9 5± 1.34 ^a^	83.90 ± 0.88 ^b^	92.41 ± 2.01 ^a^	91.01 ± 1.21 ^a^	90.93 ± 1.62 ^a^	90.04 ^AB^			
	**12**	87.14 ± 1.17 ^ab^	82.91 ± 1.05 ^b^	88.99 ± 1.84 ^a^	89.83 ± 1.45 ^a^	89.53 ± 1.85 ^a^	87.68 ^B^			
	**Mean**	90.58 ^A^	84.83 ^B^	91.62 ^A^	90.57 ^A^	90.90 ^A^				

NC = negative control, PC = positive control, FS = flaxseed, EFM = extruded flaxseed meal, FSO = flaxseed oil. *n* = 6. Means having similar letters in a row for individual parameters do not differ significantly from one another (*p* > 0.05). The lowercase letters are for one-way Anova represents weekly analysis. The uppercase letters are for two-way Anova represent the grand mean of groups and weeks.

**Table 5 foods-09-01663-t005:** Effect of different forms of flaxseed on the serum biochemical profile of Hy-line brown hens.

		Groups						*p*-Value		
	Weeks	NC	PC	FS	ESM	FSO	Mean	Groups	Weeks	Groups × Weeks
**TC (mmol/L)**	**4**	6.75 ± 0.25 ^a^	4.28 ± 0.24 ^b^	3.12 ± 0.47 ^b^	3.18 ± 0.53 ^b^	3.27 ± 0.51 ^b^	4.12 ^A^	<0.001	0.001	1.000
	**6**	6.71 ± 0.32 ^a^	4.36 ± 0.29 ^b^	3.11 ± 0.19 ^c^	3.15 ± 0.18 ^c^	3.22 ± 0.17 ^c^	4.11 ^A^			
	**8**	6.70 ± 0.38 ^a^	4.19 ± 0.36 ^b^	2.94 ± 0.16 ^c^	2.94 ± 0.16 ^c^	2.88 ± 0.13 ^c^	3.93 ^A^			
	**10**	6.13 ± 0.38 ^a^	4.21 ± 0.34 ^b^	2.87 ± 0.19 ^c^	2.84 ± 0.21 ^c^	2.68 ± 0.23 ^c^	3.75 ^AB^			
	**12**	5.97 ± 0.15 ^a^	3.74 ± 0.23 ^b^	2.57 ± 0.29 ^c^	2.35 ± 0.27 ^c^	2.29 ± 0.28 ^c^	3.39 ^B^			
	**Mean**	6.45 ^A^	4.16 ^B^	2.93 ^C^	2.90 ^C^	2.87 ^C^				
**TG (mmol/L)**	**4**	17.27 ± 0.71 ^a^	15.43 ± 0.36 ^b^	13.77 ± 0.29 ^c^	13.88 ± 0.40 ^c^	13.55 ± 0.23 ^c^	14.78 ^A^	<0.001	<0.001	0.825
	**6**	17.21 ± 1.01 ^a^	15.37 ± 0.40 ^b^	13.70 ± 0.32 ^c^	13.65 ± 0.34 ^c^	13.48 ± 0.36 ^c^	14.68 ^A^			
	**8**	16.54 ± 1.05 ^a^	14.71 ± 0.41 ^a^	12.54 ± 0.55 ^b^	12.38 ± 0.52 ^b^	12.21 ± 0.57 ^b^	13.67 ^B^			
	**10**	17.18 ± 1.65 ^a^	14.68 ± 0.31 ^b^	11.51 ± 0.51 ^c^	11.68 ± 0.44 ^c^	11.52 ± 0.34 ^c^	13.31 ^B^			
	**12**	17.45 ± 1.29 ^a^	14.46 ± 0.70 ^b^	11.45 ± 0.44 ^c^	11.62 ± 0.36 ^c^	11.29 ± 0.44 ^c^	13.26 ^B^			
	**Mean**	17.13 ^A^	14.93 ^B^	12.64 ^C^	12.60 ^C^	12.41 ^C^				
	**4**	4.38 ± 0.27 ^a^	3.13 ± 0.39 ^b^	2.88 ± 0.25 ^b^	2.71 ± 0.23 ^b^	2.65 ± 0.24 ^b^	3.15	<0.001	0.286	1.000
**LDL-C** **(mmol/L)**	**6**	4.28 ± 0.30 ^a^	3.16 ± 0.22 ^b^	2.70 ^b^ ± 0.15 ^c^	2.61 ± 0.16 ^bc^	2.45 ± 0.07 ^c^	3.04			
	**8**	4.31 ± 0.33 ^a^	3.15 ± 0.45 ^b^	2.48 ± 0.29 ^b^	2.41 ± 0.28 ^b^	2.25 ± 0.16 ^b^	2.92			
	**10**	4.25 ± 0.29 ^a^	3.13 ± 0.24 ^b^	2.41 ± 0.20 ^c^	2.33 ± 0.18 ^c^	2.17 ± 0.22 ^c^	2.86			
	**12**	4.28 ± 0.30 ^a^	3.11 ± 0.34 ^b^	2.45 ± 0.14 ^bc^	2.38 ± 0.12^c^	2.05 ± 0.17 ^c^	2.86			
	**Mean**	4.30 ^A^	3.14 ^B^	2.56^C^	2.49 ^C^	2.31 ^C^				
	**4**	1.12 ± 0.21 ^b^	2.27 ± 0.32 ^a^	2.85 ± 0.21 ^a^	2.80 ± 0.23 ^a^	2.71 ± 0.24 ^a^	2.35	<0.001	0.909	1.000
	**6**	1.21 ± 0.24 ^b^	2.27 ± 0.24 ^a^	2.76 ± 0.21^a^	2.60 ± 0.78 ^a^	2.52 ± 0.52 ^a^	2.27			
**HDL-C** **(mmol/L)**	**8**	1.06 ± 0.20 ^b^	2.23 ± 0.15 ^ab^	2.89 ± 0.41 ^a^	2.56 ± 0.08 ^a^	2.28 ± 0.18 ^a^	2.20			
	**10**	1.15 ± 0.37 ^b^	2.31 ± 0.26 ^a^	2.91 ± 0.48 ^a^	2.58 ± 0.26 ^a^	2.23 ± 0.25 ^a^	2.24			
	**12**	1.05 ± 0.21 ^c^	2.22 ± 0.23 ^b^	2.88 ± 0.23 ^a^	2.71 ± 0.14 ^ab^	2.38 ± 0.18 ^ab^	2.25			
	**Mean**	1.12 ^C^	2.26 ^B^	2.86^A^	2.65 ^AB^	2.43 ^B^				
	**4**	2.09 ± 0.22 ^a^	1.59 ± 0.11 ^b^	1.36 ± 0.05 ^bc^	1.31 ± 0.04 ^bc^	1.12 ± 0.16 ^c^	1.49	<0.001	0.668	1.000
**VLDL-C** **(mmol/L)**	**6**	2.09 ± 0.21 ^a^	1.55 ± 0.09 ^b^	1.36 ± 0.10 ^bc^	1.31 ± 0.07 ^bc^	1.11 ± 0.13 ^c^	1.48			
	**8**	1.92 ± 0.17 ^a^	1.59 ± 0.13 ^ab^	1.35 ± 0.09 ^bc^	1.19 ± 0.08 ^bc^	1.02 ± 0.16 ^c^	1.43			
	**10**	2.04 ± 0.19 ^a^	1.60 ± 0.09 ^ab^	1.41 ± 0.11 ^bc^	1.13 ± 0.20 ^bc^	0.96 ± 0.23 ^c^	1.41			
	**12**	2.02 ± 0.18 ^a^	1.51 ± 0.11 ^b^	1.36 ± 0.12 ^bc^	1.02 ± 0.23 ^bc^	0.94 ± 0.18 ^c^	1.37			
	**Mean**	2.03 ^A^	1.56 ^B^	1.37 ^AB^	1.19 ^CD^	1.03 ^D^				
**LPL (U/mL)**	**4**	442.91 ± 5.70 ^b^	787.34 ± 25.28 ^a^	815.26 ± 33.15 ^a^	821.92 ± 17.78 ^a^	815.25 ± 13.81 ^a^	736.54	<0.001	0.848	0.995
	**6**	451.99 ± 6.39 ^b^	782.33 ± 31.43 ^a^	833.92 ± 25.64 ^a^	798.59 ± 32.84 ^a^	826.92 ± 25.41 ^a^	738.75			
	**8**	460.48 ± 12.40 ^b^	805.67 ± 35.74 ^a^	798.59 ± 16.30 ^a^	808.59 ± 11.68 ^a^	831.92 ± 25.91 ^a^	741.05			
	**10**	452.67 ± 5.89 ^b^	784.00 ± 42.73 ^a^	793.59 ± 14.99 ^a^	801.92 ± 14.37 ^a^	795.26 ± 25.84 ^a^	725.48			
	**12**	450.62 ± 8.66 ^b^	805.83 ± 47.58 ^a^	831.92 ± 34.34 ^a^	825.99 ±2 0.86 ^a^	792.59 ± 20.71 ^a^	741.39			
	**Mean**	451.73 ^B^	793.03 ^A^	811.40 ^A^	812.39 ^A^	814.66 ^A^				

NC = negative control, PC = positive control, FS = flaxseed, EFM = extruded flaxseed meal, FSO = flaxseed oil. *p* < 0.05 *n* = 6 TC = total cholesterol TG = triglycerides, LDL-C = low density lipoprotein cholesterol, HDL-C = high density lipoprotein cholesterol, VLDL-C = very low density lipoprotein cholesterol, LPL = lipoprotein lipase. Means having similar letters in a row for individual parameters do not differ significantly from one another (*p* > 0.05). The lowercase letters are for one-way Anova represents weekly analysis. The uppercase letters are for two-way Anova represent the grand mean of groups and weeks.

**Table 6 foods-09-01663-t006:** Effect of different forms of Flaxseed on n-6, Tn-6, Tn-3, and n6:n3 ratio of egg yolk expressed as mg/50 g of egg.

Groups								*p*-Values		
FA	Weeks	NC	PC	FS	EFM	FSO	Mean	Groups	Weeks	Group × Weeks
**DHA**	2	56.68 ± 1.52 ^c^	83.66 ± 5.61 ^b^	92.33 ± 1.60 ^a^	96.33 ± 0.72 ^a^	97.33 ± 1.89 ^a^	85.27 ^C^	<0.001	<0.001	<0.001
	4	64.51 ± 3.84 ^d^	88.68 ± 5.44 ^c^	112.35 ± 2.14 ^b^	115.18 ± 3.18 ^b^	132.85 ± 2.91 ^a^	102.72 ^B^			
	6	58.52 ± 2.45 ^d^	92.02 ± 1.45 ^c^	125.85 ± 4.21 ^b^	126.84 ± 1.70 ^b^	139.35 ± 4.08 ^a^	108.52 ^B^			
	8	65.35 ± 1.51 ^d^	111.18 ± 2.43 ^c^	136.18 ± 3.09 ^b^	137.01 ± 6.22 ^b^	154.71 ± 5.99 ^a^	120.89 ^A^			
	10	66.18 ± 3.39 ^d^	115.01 ± 2.50 ^c^	138.85 ± 2.63 ^b^	139.18 ± 1.92 ^b^	156.01 ± 9.62 ^a^	123.05 ^A^			
	12	68.35 ± 0.60 ^d^	115.85 ± 1.97 ^c^	129.68 ± 4.69 ^b^	136.51 ± 4.82 ^b^	152.18 ± 6.74 ^a^	120.51 ^A^			
	**Mean**	63.26 ^D^	101.07 ^C^	122.54 ^B^	125.18 ^B^	138.74 ^A^				
	2	1492.44 ± 1.45 ^a^	1312.27 ± 30.20 ^b^	1358.94 ± 17.85 ^b^	1364.27 ± 28.92 ^b^	1238.94 ± 81.23 ^b^	1353.37 ^A^	<0.001	<0.001	<0.001
**Total n-6**	4	1458.72 ± 2.36 ^a^	1242.04 ± 17.44 ^c^	1314.04 ± 34.86 ^b^	1259.54 ± 3.21 ^bc^	1092.04 ± 29.73 ^d^	1273.28 ^B^			
	6	1468.89 ± 10.09 ^a^	1049.06 ± 31.69 ^c^	1201.56 ± 43.33 ^b^	1000.05 ± 18.93 ^cd^	948.89 ± 32.58 ^d^	1133.69 ^C^			
	8	1551.12 ± 3.65 ^a^	976.96 ± 31.01 ^b^	894.95 ± 22.58 ^c^	861.95 ± 2.91 ^c^	895.79 ± 20.71 ^c^	1036.16 ^D^			
	10	1534.51 ± 3.78 ^a^	941.85 ± 14.95 ^b^	862.68 ± 33.79 ^b^	775.85 ± 41.15 ^c^	880.18 ± 19.91 ^b^	999.02 ^D^			
	12	1546.18 ± 8.73 ^a^	934.68 ± 22.07 ^b^	853.68 ± 38.63 ^b^	770.85 ± 39.17 ^c^	871.18 ± 21.20 ^b^	995.32 ^D^			
	**Mean**	1508.65 ^A^	1076.14 ^B^	1080.98 ^B^	1005.42 ^C^	987.84 ^C^				
	2	85.92 ± 0.83 ^c^	268.76 ± 19.70 ^b^	341.88 ± 8.69 ^a^	354.04 ± 9.00 ^a^	370.38 ± 5.49 ^a^	284.20 ^D^	<0.001	<0.001	<0.001
	4	89.60 ± 2.18 ^c^	282.59 ± 20.02 ^b^	350.71 ± 5.57 ^a^	365.37 ± 5.90 ^a^	381.21 ± 12.13 ^a^	293.90 ^D^			
**Total n-3**	6	96.10 ± 0.75 ^c^	321.92 ± 3.41 ^b^	413.21 ± 14.43 ^a^	434.04 ± 9.09 ^a^	437.87 ± 18.69 ^a^	340.63 ^C^			
	8	104.76 ± 1.26 ^c^	385.92 ± 23.15 ^b^	457.21 ± 6.24 ^a^	466.71 ± 12.96 ^a^	474.21 ± 7.92 ^a^	377.76 ^B^			
	10	114.42 ± 1.24 ^d^	416.76 ± 5.70 ^c^	533.71 ± 5.14 ^b^	540.37 ± 3.74 ^b^	570.87 ± 5.60 ^a^	435.23 ^A^			
	12	117.76 ± 2.48 ^d^	425.10 ± 7.59 ^c^	542.04 ± 9.42 ^b^	548.71 ± 3.20 ^b^	578.21 ± 5.35 ^a^	442.36 ^A^			
	**Mean**	101.43 ^D^	350.18 ^C^	439.72 ^B^	451.54 ^B^	468.79 ^A^				
	2	17.38 ± 0.17 ^a^	5.01 ± 0.36 ^b^	3.99 ± 0.11 ^c^	3.86 ± 0.08 ^c^	3.36 ± 0.24 ^c^	6.72 ^A^	<0.001	<0.001	<0.001
	4	16.33 ± 0.41 ^a^	4.50 ± 0.32 ^b^	3.75 ± 0.08 ^c^	3.45 ± 0.05 ^cd^	2.87 ± 0.05 ^d^	6.18 ^B^			
	6	15.29 ± 0.14 ^a^	3.26 ± 0.11 ^b^	2.92 ± 0.11 ^c^	2.31 ± 0.05 ^d^	2.19 ± 0.13 ^d^	5.19 ^C^			
**n-6:n-3**	8	14.82 ± 0.20 ^a^	2.58 ± 0.18 ^ab^	1.96 ± 0.05 ^c^	1.85 ± 0.05 ^c^	1.89 ± 0.08 ^c^	4.62 ^D^			
	10	13.42 ± 0.17 ^a^	2.26 ± 0.04 ^b^	1.62 ± 0.05 ^c^	1.44 ± 0.08 ^c^	1.54 ± 0.04 ^c^	4.06 ^E^			
	12	13.16 ± 0.26 ^a^	2.21 ± 0.08 ^b^	1.58 ± 0.07 ^c^	1.41 ± 0.07 ^c^	1.51 ± 0.04 ^c^	3.97 ^E^			
**Mean**		15.06 ^A^	3.30 ^B^	2.63 ^C^	2.39 ^D^	2.23 ^D^				

NC = negative control, PC = positive control, FS = flaxseed, EFM = extruded flaxseed meal, FSO = flaxseed oil. *p* < 0.05 *n* = 6 DHA = docosahexaenoic acid. Means having similar letters in a row for individual parameters do not differ significantly from one another (*p* > 0.05). The lowercase letters are for one-way Anova represents weekly analysis. The uppercase letters are for two-way Anova represent the grand mean of groups and weeks.

**Table 7 foods-09-01663-t007:** Effect of different forms of flaxseed on the relative gene mRNA expression.

Group	Gene Expression							
	*ACOX1*	*LCPT1*	*PPAR-α*	*FADS1*	*FADS2*	*Elov2*	*Elov5*	*LPL*
**NC**	0.62 ± 0.06 ^d^	0.52 ± 0.06 ^d^	1.05 ± 0.01 ^a^	0.38 ± 0.04 ^d^	0.36 ± 0.04 ^c^	0.43 ± 0.04 ^d^	0.95 ± 0.10	0.55 ± 0.08 ^b^
**PC**	0.91 ± 0.02 ^c^	0.90 ± 0.02 ^c^	0.84 ± 0.13 ^b^	0.81 ± 0.15 ^c^	0.74 ± 0.15 ^c^	0.99 ± 0.05 ^c^	1.00 ± 0.10	0.51 ± 0.10 ^b^
**FS**	1.36 ± 0.11 ^b^	1.14 ± 0.04 ^b^	0.59 ± 0.06 ^c^	1.34 ± 0.06 ^b^	1.40 ± 0.04 ^b^	1.28 ± 0.05 ^b^	1.04 ± 0.03	1.15 ± 0.02 ^a^
**FSM**	1.28 ± 0.04 ^b^	1.22 ± 0.04 ^b^	0.60 ± 0.05 ^c^	1.45 ± 0.10 ^b^	1.55 ± 0.07 ^b^	1.39 ± 0.09 ^b^	1.04 ± 0.11	1.08 ± 0.02 ^a^
**FSO**	1.65 ± 0.05 ^a^	1.43 ± 0.05 ^a^	0.32 ± 0.03 ^d^	2.14 ± 0.12 ^a^	2.20 ± 0.10 ^a^	1.61 ± 0.08 ^a^	0.76 ± 0.08	1.18 ± 0.02 ^a^
***p*-value**	<0.001	<0.001	<0.001	<0.001	<0.001	<0.001	0.182	<0.001

NC = negative control, PC = positive control, FS = flaxseed, EFM = extruded flaxseed meal, FSO = flaxseed oil. *p* < 0.05 *n* = 6. Means having similar letters in a row for individual parameters do not differ significantly from one another (*p* > 0.05).

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
