# Peer review of "Comparative Effects of Flaxseed Sources on the Egg ALA Deposition and Hepatic Gene Expression in Hy-Line Brown Hens"

_foods, 2020, doi:10.3390/foods9111663_

Round 1

Reviewer 1 Report

The article presented is very interesting. The assumptions of the work are correct, the selection of research methods is correct and sufficient.

The prepared text is intelligible and does not raise any reservations. The obtained results were properly described with statistical analysis appropriate for the methods used.

The only reservation may be raised by 3 self-citations, but it is not a disqualifying number, especially as it proves the research team's extensive experience.

Author Response

Reviewer 1

The article presented is very interesting. The assumptions of the work are correct, the selection of research methods is correct and sufficient.

Response: Thank you so much for reviewing and appreciating our manuscript.

The prepared text is intelligible and does not raise any reservations. The obtained results were properly described with statistical analysis appropriate for the methods used.

Response: Thank you for the approval of statistical analysis.

 The only reservation may be raised by 3 self-citations, but it is not a disqualifying number, especially as it proves the research team's extensive experience.

Response: Thank you for sorting our published self-citations and appreciating our research team's experience.

Reviewer 2 Report

Comments on the paper foods-966867 submitted for publication in the journal Foods

The paper by Muhammad Suhaib Shahid and co-wrokers addressed an interesting topic aiming to investigate the efficiency of different dietary sources of n-3 on the n-3 deposition in the egg yolk and gene expression in laying hens.

Overall the paper is interesting; however major English editing is needed and the statistics are not appropriate. I suggest checking the manuscript by a native English speaker to avoid confusion in the meaning of some sentences.

The abstract is not very attractive. Rewording and major modifications are needed.

The objectives and main hypothesis are not very clear from the introduction. Further, authors are conspicuously avoiding citing interesting recent paper in the same field of this paper. Please, I recommend to further considering other studies to better situate this study and also highlight its novelty.

Further details on the experimental design are missing. Please detail as much as possible.

In table 1, the authors are using abbreviations. They need to be defined in the footnote.

The title of sub-section 2.5 needs changes.

The section 2.7 Statistical analysis is very weak. The main drawback of this study is statistics. Please, further analyses such as multivariate analyses are needed here. The reviewer suggests Principal Component Analysis, to highlight the differences among the groups. Also what about the interactions? This is mandatory and should be considered in the whole manuscript.

The authors should justify how they selected the genes given in Table 2. References are needed.

Tables 3-7, the interactions should be shown and discussed. This is very important.

Sub-sections are needed in the discussion. Please consider to include attractive titles to understand what the first announcement is and what the key home messages are. The discussion should consider important and ne references in the field.

Please remove or avoid citing all the references that has no relation to the field of this study and from other species.

Author Response

The paper by Muhammad Suhaib Shahid and co-wrokers addressed an interesting topic aiming to investigate the efficiency of different dietary sources of n-3 on the n-3 deposition in the egg yolk and gene expression in laying hens.

Response: Thank you so much for reviewing and appreciating our manuscript.

Overall the paper is interesting; however major English editing is needed and the statistics are not appropriate. I suggest checking the manuscript by a native English speaker to avoid confusion in the meaning of some sentences.

Response: English editing had done and statistical analysis modified.

The abstract is not very attractive. Rewording and major modifications are needed.

Response: Changes to abstract had done.

The objectives and main hypothesis are not very clear from the introduction. Further, authors are conspicuously avoiding citing interesting recent paper in the same field of this paper. Please, I recommend to further considering other studies to better situate this study and also highlight its novelty.

Response: Introduction was added a good back-ground and other modifications. The latest relating studies was already added to the introduction. There are not more studies regarding the flaxseed sources. That’s why these are added to this study.

Further details on the experimental design are missing. Please detail as much as possible.

Response: Details added in the material and methods sub-sections.

In table 1, the authors are using abbreviations. They need to be defined in the footnote.

  • Response: Abbreviations were defined in table 1 footnote

The title of sub-section 2.5 needs changes.

  • Response: Changes added.

The section 2.7 Statistical analysis is very weak. The main drawback of this study is statistics. Please, further analyses such as multivariate analyses are needed here. The reviewer suggests Principal Component Analysis, to highlight the differences among the groups. Also what about the interactions? This is mandatory and should be considered in the whole manuscript.

  • Response: Detail of statistical analysis was added in section 2.7.

The authors should justify how they selected the genes given in Table 2. References are needed.

  • Response: References of each gene as accession number was added in table 2.

Tables 3-7, the interactions should be shown and discussed. This is very important.

Response: The most important thing interaction was added to tables 3-7 and discussed in the result section.

Sub-sections are needed in the discussion. Please consider to include attractive titles to understand what the first announcement is and what the key home messages are. The discussion should consider important and ne references in the field.

Response: The format of Foods journal doesn’t allow us to add sub-sections to the discussion section.

Please remove or avoid citing all the references that has no relation to the field of this study and from other species.

Reviewer 3 Report

The subject of the article is interesting, but I noticed the following errors and irregularities that suggest preparing the article in a hurry:

- many extra spaces between words or no space between words (for example rows: 15, 50, selections of text in Keywords, no dots (row 102) confusing upper and lower case letters in words (for example row 58, Tab. 2) ,

- no italics of gene names (row 184 and 185, Tab. 7),

- no line numbers in the discussion,

- Table 2 should be placed in the text before line 110, ie "Statistical analysis".

Besides, in the article there is some inconsistency in the information regarding the timing of the research. If the Authors, as in row 100, give the deadline in weeks, it seems to me that the note "research" is needed. If in row 106 the Authors say that "at ten weeks (research should be given here), one bird from every replicate was slaughtered", they could not take blood from them later in week 12 (there is no note here that "research"). The number of birds used for each type of analysis must be reported.

The basic error in the statistical analysis is the one-way ANOVA analysis. If the researchers used birds in different food groups (feed composition was a factor), the cage could also be a factor (composition of 3 individuals in 1 cage). This is a hierarchical layout / analysis. Perhaps the authors checked whether there was a statistically significant group influence. If so, it is necessary to provide such information.

The article may be published only after additional statistical analyzes have been carried out and relevant information has been included in the article.

Author Response

Please see the attached point to point response to your review.

Report of reviewer #3

Comments and Suggestions for Authors

The subject of the article is interesting, but I noticed the following errors and irregularities that suggest preparing the article in a hurry:

Answer: Thank you for your review.

- many extra spaces between words or no space between words (for example rows: 15, 50,

selections of text in Keywords, no dots (row 102) confusing upper and lower case letters in

words (for example row 58, Tab. 2) ,

Answer: These mistake along with other throughout the text were corrected.

- no italics of gene names (row 184 and 185, Tab. 7),

Answer: names of genes in table 2 and throughout the text were italicized

- no line numbers in the discussion,

Answer: line numbers provided.

- Table 2 should be placed in the text before line 110, i.e. "Statistical analysis"

Answer: Table 1 and 2 both were placed before statistical analysis

Besides, in the article there is some inconsistency in the information regarding the timing of

the research. If the Authors, as in row 100, give the deadline in weeks, it seems to me that

the note "research" is needed. If in row 106 the Authors say that "at ten weeks (research

should be given here), one bird from every replicate was slaughtered", they could not take

blood from them later in week 12 (there is no note here that "research"). The number of

birds used for each type of analysis must be reported.

Answer: The Inconsistency was removed. The term “at ten weeks” was replaced with 12 weeks. Number of birds was mentioned for blood collection. Time of collection and number of birds per replicate for each analysis was mentioned.

The basic error in the statistical analysis is the one-way ANOVA analysis. If the researchers

used birds in different food groups (feed composition was a factor), the cage could also be a

factor (composition of 3 individuals in 1 cage). This is a hierarchical layout / analysis.

Perhaps the authors checked whether there was a statistically significant group influence. If

so, it is necessary to provide such information.

Answer: AS we introduced in the Material and Methods section that each replicate had fifteen hens, with separate compartments for three hens in each cage. However, five conjoint cage was a same replicate, fed with same diet. Therefore, we cannot regard the cage as a factor. This information was added in the Material and Method section.

Reviewer 4 Report

The manuscript of Shahid and colleagues aims to analyze the n-3 PUFAs deposition and hepatic gene expression after three different flaxseed sources and carbohydrates enzymes supplementations in Hy-line brown hens. However, in my opinion, several improvements are necessary before publication. You can find some suggestions below.

  • Title: considering that the sources are all from flaxseed, I believe that ‘flaxseed sources’ it would be more appropriate.
  • Because I can find several English imprecisions I suggest English revision.
  • When you talk about ‘n-3’, you must always add ‘PUFA’.
  • Considering the large difference on health among precursor and effectors of n-3 PUFAs, I suggest to use the type of n-3 in question (e.g. ALA or DHA and so on) rather than the generic ‘n-3’, in the title as throughout the text.
  • Please define all the acronyms, and only the first time you mention them, throughout the abstract, text, and figures/tables (e.g. HDEP, FCR).
  • “Linolenic acid or n-3 (C18:3n3) is helpful to regress atherosclerosis and improve brain health” this sentence contains two inaccuracies: first one, ALA is an n-3, but putting ‘or’ it seems that is the only one, it would be better to define, for example ‘as one of the omega-3’. Second one, omega-3 have numerous health benefits, so in this way is very reductive.
  • Line 37: ‘omega-3’: as ‘n-3’ and ‘omega-3’ indicate the same group, you need to standardize the nomenclature, always one of them.
  • Lines 76-77: I really don’t understand why the corn-based diet was considered as ‘positive’ control, even if corn is rich of n-6 fatty acids than n-3 (as reported in table 1). I suggest to change the name or give some strong explanation in the text. Moreover, in this regard, how the data for ‘nutrient analysis’ in table 1 were obtain? You must add information in material and methods and means/SE in the table.
  • Line 80-81: some brief information about the production conditions for the used EFM would be necessary.
  • Lines 82-83: “the diets were formulated to be iso-caloric and iso-nitrogenous” please include these information in table 1.
  • Lines 86: please explain the difference and modality among the types of measurement.
  • Line 106: please add information about the method of anesthesia and sacrifice, and the details of approval by a properly constituted research ethics committee (as reported in the Foods instructions for authors: as a minimum, the project identification code, date of approval and name of the ethics committee or institutional review board should be cited in the Methods section).
  • Line 111: ‘one-way’ replace zero with the letter o.
  • Table 2: add the explanation of all acronyms in the note.
  • Table 3-4-5-6-7: in my opinion SE values for all the means, letters for all the means (even if not statistically significant), explanation of the meaning of the letters in each table in the notes should be added. Moreover, please add horizontal lines to separate data among each group.
  • Please explain somewhere how FCR, albumen weight, and Haugh unit data were obtained.
  • Please move each tables near to the text description.
  • Line 121: ‘wks’ please correct it.
  • Line 152: “The n-3 sources of flaxseed affected serum lipid profile” Because the meaning of the letters is not explained and SE values for each means are not shown, I supposed that means with different letters are significantly different. In that case when two means share the same letter they are not statistically significant, and this is the case for the results of table 5 for all the experimental groups FS, EFM, and FSO, so this sentence is wrong.
  • Lines 178 and 180: ‘Tn-3’ please correct it.
  • Table 7: the name of the groups it should be like all the previous tables.
  • Other misspelled words in the discussion (no line numbers in the discussion for the correct indication): FS instead of flaxseed, ‘our’, and ‘Competition’.

Author Response

The manuscript of Shahid and colleagues aims to analyze the n-3 PUFAs deposition and hepatic gene expression after three different flaxseed sources and carbohydrates enzymes supplementations in Hy-line brown hens. However, in my opinion, several improvements are necessary before publication. You can find some suggestions below.

Response: Thank you so much for your review and suggestions.

  • Title: considering that the sources are all from flaxseed, I believe that ‘flaxseed sources’ it would be more appropriate.
  • Response: ‘flaxseed sources’ added to the title
  • Because I can find several English imprecisions I suggest English revision.
  • Response: English editing has been done
  • When you talk about ‘n-3’, you must always add ‘PUFA’.
  • Response: Added
  • Considering the large difference on health among precursor and effectors of n-3 PUFAs, I suggest to use the type of n-3 in question (e.g. ALA or DHA and so on) rather than the generic ‘n-3’, in the title as throughout the text.
  • Response: agreed and modified.
  • Please define all the acronyms, and only the first time you mention them, throughout the abstract, text, and figures/tables (e.g. HDEP, FCR).
  • Response: Acronyms defined.
  • “Linolenic acid or n-3 (C18:3n3) is helpful to regress atherosclerosis and improve brain health” this sentence contains two inaccuracies: first one, ALA is an n-3, but putting ‘or’ it seems that is the only one, it would be better to define, for example ‘as one of the omega-3’. Second one, omega-3 have numerous health benefits, so in this way is very reductive.
  • Response: Sentences modified and defined.
  • Line 37: ‘omega-3’: as ‘n-3’ and ‘omega-3’ indicate the same group, you need to standardize the nomenclature, always one of them.
  • Response: Nomenclature standardized throughout the text.
  • Lines 76-77: I really don’t understand why the corn-based diet was considered as ‘positive’ control, even if corn is rich of n-6 fatty acids than n-3 (as reported in table 1). I suggest to change the name or give some strong explanation in the text. Moreover, in this regard, how the data for ‘nutrient analysis’ in table 1 were obtain? You must add information in material and methods and means/SE in the table.
  • Response: sorry I did a mistake. You are right the PC must be NC and vice versa. I had changed the name of PC with NC and NC with PC.
  • Line 80-81: some brief information about the production conditions for the used EFM would be necessary.
  • Response: The EFM was imported from Canada and the producers did not reply to our email. Sorry for this.
  • Lines 82-83: “the diets were formulated to be iso-caloric and iso-nitrogenous” please include these information in table 1.
  • Response: the diets were formulated iso-caloric and is-onitrogenous and these information were already listed in table 1.
  • Lines 86: please explain the difference and modality among the types of measurement.
  • Response: explanation added in some sub sections of material and methods
  • Line 106: please add information about the method of anesthesia and sacrifice, and the details of approval by a properly constituted research ethics committee (as reported in the Foods instructions for authors: as a minimum, the project identification code, date of approval and name of the ethics committee or institutional review board should be cited in the Methods section).
  • Response: Method of sacrifice, was added in subsection 2.6. Ethical committee details were also added in the sub section 2.1.
  • Line 111: ‘one-way’ replace zero with the letter o.
  • Response: Replaced.
  • Table 2: add the explanation of all acronyms in the note.
  • Response: Acronyms explanation added to Table 2.
  • Table 3-4-5-6-7: in my opinion SE values for all the means, letters for all the means (even if not statistically significant), explanation of the meaning of the letters in each table in the notes should be added. Moreover, please add horizontal lines to separate data among each group.
  • Response: SE values were added to all tables.
  • Please explain somewhere how FCR, albumen weight, and Haugh unit data were obtained.
  • Response: FCR detail was added in section 2.3. while albumen height and Haugh unit detail was added in section 2.4.
  • Please move each tables near to the text description.
  • Response: Moved each table near to the text description.
  • Line 121: ‘wks’ please correct it.
  • Response: corrected.
  • Line 152: “The n-3 sources of flaxseed affected serum lipid profile” Because the meaning of the letters is not explained and SE values for each means are not shown, I supposed that means with different letters are significantly different. In that case when two means share the same letter they are not statistically significant, and this is the case for the results of table 5 for all the experimental groups FS, EFM, and FSO, so this sentence is wrong.
  • Response: sentence corrected.
  • Lines 178 and 180: ‘Tn-3’ please correct it.
  • Response: corrected.
  • Table 7: the name of the groups it should be like all the previous tables.
  • Response: Names corrected.
  • Other misspelled words in the discussion (no line numbers in the discussion for the correct indication): FS instead of flaxseed, ‘our’, and ‘Competition’.
  • Response: corrected.

Round 2

Reviewer 2 Report

The authors addressed all my comments. 

Reviewer 4 Report

The authors addressed all my comments.